# Primary Graft Dysfunction in Lung Transplantation: An Overview of the Molecular Mechanisms

**DOI:** 10.3390/ijms26146776

**Published:** 2025-07-15

**Authors:** Jitte Jennekens, Sue A. Braithwaite, Bart Luijk, Niels P. van der Kaaij, Nienke Vrisekoop, Saskia C. A. de Jager, Linda M. de Heer

**Affiliations:** 1Department of Cardiothoracic Surgery, University Medical Center Utrecht, Heidelberglaan 100, 3584 CX Utrecht, The Netherlands; j.jennekens-4@umcutrecht.nl (J.J.); n.p.vanderkaaij-2@umcutrecht.nl (N.P.v.d.K.); 2Department of Anesthesiology, University Medical Center Utrecht, Heidelberglaan 100, 3584 CX Utrecht, The Netherlands; s.a.braithwaite@umcutrecht.nl; 3Department of Pulmonology, University Medical Center Utrecht, Heidelberglaan 100, 3584 CX Utrecht, The Netherlands; h.d.luijk@umcutrecht.nl; 4Center for Translational Immunology (CTI), University Medical Center Utrecht, Heidelberglaan 100, 3584 CX Utrecht, The Netherlands; n.vrisekoop@umcutrecht.nl; 5Experimental Cardiology Laboratory, Department of Cardiology, University Medical Center Utrecht, Heidelberglaan 100, 3584 CX Utrecht, The Netherlands; s.c.a.dejager@umcutrecht.nl

**Keywords:** primary graft dysfunction, ischemia reperfusion injury, lung transplantation, oxidative stress, inflammation, neutrophils

## Abstract

Primary graft dysfunction (PGD) remains a major complication after lung transplantation. Donor lung ischemia followed by reperfusion drives oxidative stress and inflammatory responses. The pathophysiology is influenced by various donor-, procedure-, and recipient-related factors, which complicates the identification of biomarkers for evaluation of donor lung injury or therapeutic interventions to minimize PGD. This review provides an overview of the molecular pathways that contribute to PGD pathophysiology, including those involved in loss of endothelial–epithelial membrane integrity, neutrophil infiltration, and the development of pulmonary edema.

## 1. Introduction

Primary graft dysfunction (PGD) is a major cause of early morbidity and mortality following lung transplantation. It refers to any functional and/or morphological impairment of lung allografts in the first 72 h following lung transplantation. It is diagnosed and graded according to the 2005 International Society for Heart and Lung Transplantation Primary Graft Dysfunction Definition with grades ranging from zero to the highest grade three, assessed at four timepoints every 24 h (Table 1) [1]. PGD occurs after approximately 30% of lung transplantation procedures, with a reported incidence of severe-type grade 3 PGD (PGD3) at 48 and 72 h as high as 15–20% [2]. PGD is a multifactorial syndrome and its development is influenced by a complex interplay of donor-, procedure-, and recipient-related factors. These factors drive oxidative stress and inflammatory responses through molecular pathways activated before, during, and after transplantation [3]. This review comprehensively outlines the clinical, pathophysiological, and molecular contributors to the development of PGD following lung transplantation. We suggest that with a deeper understanding of the molecular mechanisms driving PGD, strategies for improved risk assessment and therapeutic interventions to prevent or treat PGD, targeted to specific donor allografts, recipients, and/or specific high-PGD risk donor–recipient combinations, could be developed.

## 2. Pathophysiologic Hallmarks of PGD

PGD clinically presents with impaired oxygenation and increased pulmonary vascular resistance (PVR). Loss of lung function can be extreme and may even lead to the need for extracorporeal life support in order to support oxygenation and ventilation. The decreased function of donor lungs can be primarily attributed to pulmonary edema and subsequent shunting of pulmonary blood flow. This edema is caused by alveolar flooding due to a compromised endothelial–epithelial barrier and the loss of vascular integrity. Endothelial leak is caused by either an inflammatory insult (inflammatory type PGD) or increased hydrostatic pressure driving fluid into the alveoli (hydrostatic PGD) [4]. Inflammatory-type PGD is associated with a poorer prognosis and results in a longer ventilation time, an increased length of stay in the ICU, and poorer survival post-transplant compared to hydrostatic PGD [5].

Inflammatory-type PGD is mainly driven by lung ischemia–reperfusion injury (LIRI). At the cellular level, this is characterized by the infiltration of neutrophils into the lung graft, endothelial and epithelial cell death, and excessive inflammation. Upon reperfusion, the recipient’s innate immune response becomes activated and contributes to endothelial damage and subsequent vascular leak characterizing PGD pathophysiology [6,7,8]. Over the last few decades, it has become increasingly clear which cells and signaling mediators are involved in LIRI [9,10]. Ischemia–reperfusion is unavoidable in any solid organ transplantation, and pathways underlying organ injury likely overlap. However, as donor lungs are preserved in an inflated state with an intra-alveolar oxygen supply, the nature of injury and underlying mechanisms could differ from other solid organs, which are mostly preserved in an ischemic state in the case of static cold preservation. Throughout the transplant continuum of donor injury, death, procurement, storage, and reperfusion in the recipient, the accumulation of multiple injurious events contributes to and may amplify the inflammation occurring in the lung allograft following reperfusion. One clinically relevant marker of such lung injury incurred during the processes up until procurement may be the weight of the donor lungs, likely reflecting edema and immune infiltrate. Indeed, a recent study has shown that donor lung weight after retrieval is associated with the development of PGD [11]. Similarly, lung weight gain after 1 h of ex vivo lung perfusion (EVLP) was negatively associated with transplant suitability [12]. Lung weight may therefore be used as a proxy which can reflect the state of endothelial–epithelial membrane integrity.

## 3. Sources and Mechanisms of Oxidative Stress in LIRI

### 3.1. Cellular and Molecular Origins of Oxidative Stress

Oxidative stress is a central mechanism contributing to LIRI and the subsequent development of PGD. High levels of reactive oxygen species (ROS) are produced in the initial phase after transplantation and not only damage cellular macromolecules, but also directly affect downstream signaling pathways. When ROS are produced in excess, it overwhelms the antioxidant defense system, resulting in lipid peroxidation, protein dysfunction, DNA damage, and ultimately cell death with ensuing inflammation [13]. ROS are generated through multiple pathways, which can generally be divided into mitochondrial-dependent and mitochondrial-independent mechanisms. The specific contribution of cell types and mechanisms that drive ROS production during LIRI have not been completely elucidated and potentially vary due to the heterogenic nature of the transplant process. Different cell types in the lung allograft and the recipient’s immune system contribute to the generation of ROS, including endothelial cells (ECs), alveolar macrophages (AMs), and neutrophils [13]. Neutrophils are a prominent source of ROS through NADPH oxidase-mediated pathways due, in part, to their large numbers present in the systemic circulation and sequestration in the lung microvasculature during inflammation [14,15,16]. NADPH oxidase activity is also an important source of ROS production in ECs. This enzyme can be activated by various stimuli such as cytokines and shear stress [17]. Other enzymatic sources of ROS that have been investigated in the context of LIRI include xanthine oxidases, nitric oxide synthases (NOSs), and mitochondrial respiratory chain enzymes, as shown in Figure 1 [18].

### 3.2. Mitochondrial Mechanisms of Oxidative Stress

Ischemia results in tissue hypoxia when oxygen supply is disrupted and the metabolic demand can no longer be met. A lack of oxygen leads to the depletion of adenosine triphosphate (ATP) and a switch from aerobic respiration to anaerobic metabolism with harmful cellular effects. The rate at which ATP is depleted during ischemic organ preservation is highly temperature-dependent. ATP depletion affects ATPase-dependent ion transport across the membrane, causing intracellular accumulation of sodium (Na^+^) and calcium (Ca^2+^). Consequently, water diffuses into the cell, which causes cell swelling, rupture, and cell death. Whether this plays a significant role in oxidative stress during the specific injury incurred in the context of LIRI can be debated. Standard lung preservation involves cold ischemic storage with the lung in a semi-inflated state with a source of intra-alveolar oxygen [19]. Unlike anoxic ischemia in other solid organs preserved for transplantation, the presence of oxygen in donor lungs prevents ATP depletion [20,21]. A study by Kinnula et al. showed that xanthine oxidase activity in healthy human lungs is low, and plasma samples of lung transplant recipients suggested a minimal role for this mechanism in post-transplant complications [22]. In contrast, pre-clinical animal models have shown that inhibition of xanthine oxidase is able to reduce LIRI [23,24,25]. These models used warm ischemic periods (37 °C) varying from 90 min to 3 h with inflated lungs. Findings from such animal models for LIRI should be interpreted with care in the context of clinical lung transplantation, as the extended periods of warm ischemia not usually encountered in clinical lung transplantation could lead to the overestimation of the consequences of ischemia in clinical PGD etiology, such as succinate accumulation and xanthine oxidase-driven ROS generation. Most animal LIRI models use extended periods of warm ischemia and they vary widely regarding ischemia strategy (partial vs. complete hilar clamping), lung inflation state, ischemia and reperfusion duration, and animal species [26].

### 3.3. Mechanotransduction-Induced Oxidative Stress

A loss of shear stress in the pulmonary (micro) circulation during lung preservation alters mechanotransduction and downstream biological signaling, resulting in the generation of ROS that is not associated with anoxia [27]. The absence of pulmonary flow leads to the closure of potassium (K^+^) channels and subsequent endothelial membrane depolarization [28,29]. This activates NADPH oxidase, which catalyzes the formation of superoxide [30]. A rat model comparing ventilated ischemia to anoxia found no effect of a xanthine oxidase inhibitor in ventilated ischemic lungs and that the generation of oxygen radicals remained present, suggesting that other enzymatic mechanisms are involved in the generation of ROS [31]. In NADPH oxidase knock-out mice, ROS production did not occur in the ischemic pulmonary endothelium, suggesting that NADPH oxidase activation is an important ROS-producing mechanism in the ischemic, but not anoxic, lung [30]. Increased intracellular Ca^2+^ independent of anoxia has also been observed with cessation of pulmonary flow, mediated by voltage-gated Ca^2+^ channels in response to membrane depolarization [32]. As a downstream effect, nitric oxide (NO) is produced by endothelial NOS (eNOS), a Ca^2+^-dependent enzyme [33,34]. NO readily reacts with ROS, forming peroxynitrate and other reactive nitrogen species (RNS) [35]. Peroxynitrate is an important mediator of cellular and tissue injury due to its high reactivity with a broad range of molecules [36]. Other isoforms of NOS, including neuronal (nNOS), inducible (iNOS), and mitochondrial (mtNOS), in the context of LIRI are not completely understood. In contrast to eNOS which is constantly expressed in the vascular endothelium, iNOS requires de novo protein synthesis and is upregulated during inflammatory conditions. iNOS can be expressed in various cell types including AMs, neutrophils, platelets, and ECs. iNOS can generate far greater levels of NO in a Ca^2+^-independent manner compared to eNOS [37,38,39]. While NO has protective effects during normal physiologic conditions, high levels in the presence of ROS result in increased NO-dependent oxidative stress. It was shown that LIRI in a rabbit model induced iNOS expression, and its inhibition prevented an increase in total NOS activity. Additionally, LIRI was associated with platelet adhesion, which was attenuated by treatment with an iNOS inhibitor. This suggests that NO overproduction due to increased iNOS activity leads to RNS formation and subsequent platelet activation. In this model, platelet adhesion increased only after 2 h of warm ischemia but not after 1 h [40]. A more clinically relevant rat transplant model in which left lungs were stored for 3 h at 4 °C prior to transplant also showed upregulation of iNOS expression. Inhibition of iNOS improved post-transplant pulmonary and vascular function. The authors suggested neutrophils to be the potential producers of iNOS in LIRI [41]. These insights emphasize that oxidative stress in LIRI is not solely due to anoxic ATP depletion. Therefore, clinical lung preservation techniques should not only focus on oxygenation and cold storage but also consider the effect of mechanical factors and flow cessation on endothelial biology. Improvements in preservation solutions and perfusion protocols for EVLP are promising strategies addressing endothelial preservation.

## 4. Impact of Donor-Related Factors on Lung Injury

### 4.1. Donor Lung Preservation and Impact on LIRI

Although ROS production is observed during LIRI, the question of how oxidative stress is regulated and which pathways are involved has not been completely answered. Based on previous studies, it was demonstrated that ATP levels remained stable in lungs preserved at 10 °C for 24 h and inflated with room air, with active metabolism indicated by a decline in oxygen concentration. At 22 °C, oxygen levels decreased by 96% after 12 h, associated with a significant reduction in ATP. Interestingly, lungs preserved at 1 °C showed a 14% decrease in ATP despite lower oxygen consumption compared to preservation at 10 °C. The authors suggest that the effect could be due to temperature-dependent enzyme dysfunction within the ATP production pathway [42]. In addition, it was demonstrated in an isolated rat lung perfusion model that lung energy status (ATP content and ATP/ADP ratio) and metabolic parameters (lactate production and lactate/pyruvate-ratio) were not significantly altered until alveolar partial oxygen pressure (pO_2_) decreased to 7 mmHg or less [20]. In clinical lung transplantation, alveolar oxygen tension during static cold preservation is thus likely to be sufficient to maintain ATP levels. Nonetheless, not every donor lung is of optimal quality, and regions of atelectasis or lung edema can impair adequate preservation and contribute to anoxic ischemia-related injury. Moreover, the shift in the lung transplant field from donor lung preservation on ice to temperature-controlled preservation (~8–10 °C) affects underlying mechanisms, with increasing evidence favoring temperature-controlled storage [43,44,45,46,47]. The development of precision preservation techniques tailored to individual donor lung quality, regional injury patterns, and recipient characteristics may help to reduce LIRI and improve lung transplantation outcomes.

### 4.2. The Role of Donor Type in LIRI

There are periods within the transplantation continuum during which the graft is exposed to hypoxic conditions. Donation after circulatory death (DCD) is marked by a period of warm ischemia and hypoxia prior to cold perfusion and retrieval of the lungs, which could initiate oxidative stress responses and tissue injury through energy-dependent mechanisms. This insult exacerbates subsequent injury during cold storage and reperfusion. Moreover, during surgical implantation, the lungs are deflated and start to rewarm, marking another period of anoxic warm ischemia. A recent study showed that the core temperature of the lung rapidly increases during implantation. At 30 min after the start of implantation, lungs previously stored on ice or preserved with the LUNGguard (8 °C) increased their core temperature to 22.0 ± 4.4 °C and 24.0 ± 3.6 °C, respectively [46]. Studies in other solid organs have shown that mitochondrial dysfunction is related to the duration of warm ischemia. A rat DCD liver transplant model showed that warm ischemia duration is directly proportional to the extent of hepatic mitochondrial damage [48]. Similarly, renal warm ischemia of 20 min leads to renal mitochondrial injury in a rat model [49]. A study with donation after brain death (DBD) and DCD human hearts demonstrated cardiac stress and mitochondrial dysfunction in both donor types. However, DCD hearts suffered more cardiac stress and impaired oxidative phosphorylation compared to DBD hearts and the severity correlated with a longer warm ischemic time and subsequent cold storage time [50]. In vitro analysis of mitochondria isolated from lungs indicated that 30 min of warm ischemia did not impair mitochondrial respiration, whereas periods exceeding 45 min were associated with mitochondrial dysfunction [51]. In contrast, a rat hilar clamp model, which involves temporary disruption of pulmonary blood and airflow by placing a clamp on the lung hilum, did show that 30 min of warm ischemia with subsequent reperfusion for 60 min induced mitochondrial respiratory chain dysfunction and impaired mitochondrial viability and integrity [52]. These findings underscore the importance of minimizing warm ischemia, especially in hypoxic conditions, which is particularly relevant for marginal donor lungs, as these lungs are more vulnerable and at risk of developing PGD. With regard to DBD lung transplantation, the global inflammatory response associated with brain death has been implicated as a potential contributor to PGD pathophysiology. This can activate or damage the pulmonary endothelium, thereby exacerbating LIRI [53,54]. Transcriptomic analysis of human DBD and DCD lung tissue biopsies collected at the end of cold ischemic preservation revealed distinct signatures. Lungs from DBD donors showed activation of inflammatory pathways, whereas cell death pathways were predominantly activated in lungs from DCD donors. Among the upregulated pathways in DBD lungs, several pathways were specific for marginal DBD lungs with an indication for assessment with EVLP (defined as arterial oxygen tension (PaO_2_)/fraction of inspired oxygen (FiO_2_) < 300 mmHg and presence of pulmonary oedema or infiltrates on chest X-ray). iNOS signaling was activated in marginal DBD lungs compared to both DCD lungs and DBD lungs without an indication for EVLP, suggesting that this mechanism could contribute to brain death-related lung injury prior to retrieval [55]. Lung transplantation following DBD or DCD may lead to activation of different mechanisms of injury. Analysis of a large cohort showed no significant difference in overall survival between DBD and DCD lung transplants. However, DCD recipients were more likely to be reintubated and require extracorporeal membrane oxygenation (ECMO) within 72 h [56]. A multicenter analysis comparing DBD and DCD lung transplant outcomes reported higher rates of PGD at 0 and 24 h after transplant with DCD lungs. However, no significant difference in PGD incidence at 48 or 72 h was observed [57].

## 5. The Innate Immune Response in LIRI

### 5.1. DAMPs and PRRs Initiate the Innate Immune Response

LIRI elicits a sterile inflammatory response, and excessive inflammation is a major driving force underlying severe PGD in synergy with oxidative stress [58]. After cellular damage or death due to LIRI, damage-associated molecular patterns (DAMPs) are released in the lung allograft. These DAMPs, such as high-mobility group box 1 (HMGB1), extracellular ATP (eATP), and mitochondrial DNA (mtDNA), are recognized by pattern recognition receptors (PRRs), expressed on both immune cells, including AMs and neutrophils, as well as endothelial and epithelial cells of the lung. As a result, PRRs initiate downstream pro-inflammatory and procoagulant signaling, making them a crucial molecular gateway to the sterile inflammatory response [59]. A schematic representation of innate immune activation is shown in Figure 2.

Different PRRs exist, such as toll-like receptors (TLRs), NOD-like receptors (NLRs), and the receptors for advanced glycation end products (RAGE). In the context of LIRI, the toll-like receptors TLR2, TLR4, and TLR9 have been studied in pre-clinical models, with particular focus on TLR4 [60,61,62,63,64,65]. In a murine model of LIRI, TLR4-deficient mice showed a role for TLR4 signaling in generating early inflammatory responses. The reduction in inflammation in the absence of TLR4 could be attributed to blunted MAPK/NF-κB activation, reduced actin cytoskeletal rearrangement and gap formation in pulmonary microvascular endothelial monolayers, decreased expression of cytokines and chemokines, and a lack of neutrophil infiltration [66]. However, the specific cellular site for TLR4 activation was not investigated. Merry et al. showed that TLR4 knockdown by short interference RNA in AMs in a rat LIRI model resulted in almost complete attenuation of lung injury [67]. Another study similarly emphasized AMs as the major cell type responsible for sensing damage and amplifying the inflammatory response through TLR4 signaling during LIRI [68]. While pulmonary ECs also express TLR4, these data indicate that AMs are the more likely contributors to the initiation of inflammation via this mechanism.

The NOD-like receptor protein 3 (NLRP3) inflammasome, a cytosolic multiprotein complex that assembles in response to stress signals, belongs to the PRRs. It results in the release of IL-1β and IL-18 and can induce pyroptotic cell death associated with the release of extracellular DNA. During LIRI, NLRP3-related inflammatory responses are primarily mediated by innate immune cells, such as AMs [69]. Inflammasome activity in LIRI has been confirmed in murine hilar clamp models. Pretreatment of mice with a selective NLRP3 inflammasome inhibitor alleviated lung injury, significantly reduced IL-1β and IL-18 cytokine production, and decreased neutrophil infiltration [70]. Cytokine measurement during EVLP of human donor lungs has shown increased expression of IL-1β, demonstrating NLRP3 activity during reperfusion in the absence of a recipient’s circulating immune system [71,72,73]. In human lungs rejected for transplantation, IL-1β and TNF-α levels were significantly higher compared to clinical human lungs [74].

The membrane-bound receptor for advanced glycation end products (mRAGE) is a PRR with particular relevance to LIRI since it is constitutively highly expressed in alveolar epithelial cells [75,76]. RAGE expression was found to be upregulated in patients with acute respiratory distress syndrome (ARDS) and PGD, and while it can bind to a variety of ligands including advanced glycation end products, it has the highest binding affinity for HMGB1 [75,76]. The interaction between RAGE and HMGB1 triggers pro-inflammatory signaling cascades through activation of the NF-κB transcription factor. Evaluation of HMGB1 levels during human EVLP showed a correlation between the development of PGD and an increase in HMGB1 levels [77]. In an experimental model, treatment with an anti-HMGB1 monoclonal antibody to neutralize HMGB1 attenuated lung injury and improved graft function, suggesting that this DAMP is involved in LIRI [78]. Similarly, in a mouse hilar clamp model, blockade of RAGE and thus its signaling, was shown to attenuate LIRI [79].

Multiple signaling pathways can trigger activation of the NF-κB transcription factor, reflecting its central role in further upregulating inflammatory genes [80]. It is activated downstream of TLRs, RAGE, inflammasome, ROS, cytokines, and endoplasmic reticulum (ER) stress. When activated, NF-κB translocates to the nucleus and induces transcription of various cytokines (e.g., IL-6, TNF-α), chemokines (e.g., IL-8), and adhesion molecules (e.g., ICAM-1, VCAM-1). NF-κB also contributes to coagulation by downregulating anticoagulant proteins, linking inflammation with thrombotic complications in PGD [81].

### 5.2. Neutrophils as Mediators of PGD

Recipient neutrophil recruitment to the transplanted lung is a hallmark of PGD pathophysiology. In the early LIRI response, current evidence identifies donor-derived AMs and non-classical monocytes (NCMs) as the principal initiators of neutrophil recruitment [82,83]. Kurihara et al. showed that NLRP3 inflammasome activation in donor NCMs and subsequent production of IL-1β was required for the activation of donor AMs in a murine lung transplant model [84]. Upon activation through the IL-1 receptor and induction of the NF-κB pathway, AMs were shown to be the predominant source of CCL2. This chemokine was necessary for the recruitment of recipient classical monocytes (CMs) and contributed to the development of PGD [84]. CMs mediate neutrophil extravasation into the interstitial space via permeabilization of the pulmonary endothelium by downregulating the expression of the tight junction-associated protein ZO-2 [84,85]. NCMs have been shown to recruit neutrophils through activation of TLR signaling and the downstream release of CXCL2. Depletion of NCMs abrogated neutrophil influx in the lung [86]. Pharmacologic or genetic inhibition of NLRP3 prevented the release of IL-1β and the activation of AMs, which prevented the recruitment of CMs and the development of PGD. IL-8 is another potent neutrophil chemoattractant secreted by activated AMs or ECs, and could have diagnostic potential in the context of PGD. Lung tissue samples collected after transplantation showed that IL-8 rapidly increased after reperfusion and IL-8 levels at 2 h of reperfusion correlated with lung function, mean airway pressure, and ICU stay [87]. Perfusate analysis of 16 human EVLPs demonstrated that a logistic regression model combining IL-8 and IL-1β levels after 2 h of perfusion could separate transplanted lungs from declined lungs [88]. It is clear that both IL-8 and IL-1β are key mediators of neutrophil recruitment. However, whether they function redundantly, act sequentially or in synergy, or one plays a more dominant role in driving neutrophil infiltration during LIRI is not yet fully understood.

Histological analysis of LIRI is characterized by neutrophilic infiltration occurring within just a few hours of reperfusion. Neutrophil migration across the pulmonary endothelium involves firm adhesion to ICAM-1 expressed on ECs. Levels of soluble ICAM-1, VCAM-1, and E selectin in EVLP perfusate of human lungs were significantly higher in the PGD3 group [89]. Neutrophils induce tissue damage by releasing high amounts of ROS via NADPH oxidase pathways, releasing proteases such as neutrophil elastase and forming neutrophil extracellular traps (NETs) [90]. These NETs, composed of extracellular DNA and proteins including histones, injure tissue and promote thrombosis [91]. The key effector functions of neutrophils in LIRI make them an interesting therapeutic target. Researchers have focused on interfering with both the recruitment of neutrophils as well as the mitigation of their damaging effects. Neutrophil rolling along the pulmonary vascular wall relies on the combination of L-selectin on neutrophils and P-selectin and E-selectin on endothelial cells. Inhibition of P-selectin and E-selectin in a sheep autologous lung transplant model improved lung function and reduced neutrophil activation [92]. In addition, inhibition of neutrophil elastase in a canine single-lung transplant model ameliorated LIRI [93]. Finally, Sayah et al. showed that disruption of NETs by intrabronchial administration of DNaseI reduced lung injury in a murine PGD model [94].

### 5.3. Complement Activation and Amplification of Lung Injury

The complement cascade is an important part of the innate immune response, and its activation has been shown to contribute to ischemia–reperfusion injury (IRI) in solid organ transplantation [95,96,97,98]. Upon activation, complement can have direct and indirect effects. Fragments that are generated, such as C5a, have chemoattractant properties, promoting the recruitment of neutrophils and monocytes. In addition, it functions as a bridge between innate and adaptive immunity by providing important signals for T and B cell activation. A direct effect of complement activation is the formation of membrane attack complexes (MACs) on ECs. This results in downstream activation of the NF-κB transcription factor, the production of pro-inflammatory cytokines, and the upregulation of adhesion molecules, facilitating neutrophil infiltration and platelet aggregation [99]. Post-transplant plasma levels of complement fragments C5a and C4a were found to be elevated in lung transplant recipients with PGD [100]. Furthermore, higher levels of certain complement components in bronchoalveolar lavage (BAL) were found to be associated with PGD [101]. It has been suggested that brain death can immunologically prime a donor organ, leading to exacerbated effects of IRI [102,103]. In a mouse lung transplant model, DBD donor pretreatment with a C3a receptor antagonist significantly reduced LIRI compared to vehicle-treated DBD donors. Complement activation in brain death donors has received previous attention, and this study supports the hypothesis that pre-donation complement activation aggravates LIRI [104].

### 5.4. Recipient-Related Factors in LIRI

Processes underlying LIRI can be exacerbated by recipient-related factors, predisposing recipients to increased PGD risk. Figure 3 summarizes the donor, procedure, and recipient factors that could influence PGD pathophysiology. The use of cardiopulmonary bypass (CPB) has been identified as a risk factor for PGD, with prolonged CPB duration associated with more severe PGD and reduced survival [105,106]. This can be explained by an increase in cell-free hemoglobin (CFH) levels due to hemolysis associated with CPB use [107]. In addition, transfusion with red blood cells (RBCs), which is also known to introduce hemolytic products into a patient, can also contribute to circulating CFH. Higher levels of perioperative CFH were independently associated with increased PGD risk [108]. CFH has been shown to induce injury through scavenging of NO, pro-inflammatory signaling, and oxidative injury in patients with sepsis and sickle cell disease. However, the specific role of CFH in PGD pathophysiology is unclear. Isolated perfused human lungs exposed to intravascular CFH developed increased vascular permeability as measured by lung weight and extravasation of Evans blue-labeled albumin dye into the alveolar space [108]. The authors suggested that oxidative injury due to CFH is responsible for increased endothelial permeability. In another study seeking to understand the mechanisms through which CFH induces endothelial barrier dysfunction, it was hypothesized that cell death may be a major contributor. However, the authors showed that CFH-mediated microvascular permeability was not caused by known cell death pathways including apoptosis, necrosis, necroptosis, ferroptosis, and autophagy [109]. Nonetheless, this study did show that endothelial barrier dysfunction can be prevented by a hemoglobin scavenger, underscoring their potential role in PGD (112). Researchers have also investigated genetic predisposition in recipients as a risk factor for PGD. Somers et al. identified IL-17R genetic variants associated with higher PGD grades after lung transplant [110]. Similarly, in a large multicenter cohort study, two genetic variants in the prostaglandin E_2_ (PGE2) family were found to be significantly associated with PGD [111]. PGE2 is involved in various physiologic processes, including inflammation. The functional consequence of the two genetic variants in PGD pathophysiology is not completely understood. One genetic variant was associated with decreased Treg suppressor cell function. The same research group identified the involvement of genetic variants in the gene encoding PTX3 in PGD pathophysiology [112]. This protein is involved in the innate immune response. However, the mechanism by which certain variants increase recipient susceptibility remains elusive. It was previously shown that elevated post-transplant plasma levels of PTX3 in idiopathic pulmonary fibrosis (IPF) patients are associated with PGD, suggesting a potential functional link between genetic variants, PTX3 protein level, and PGD development [113]. Furthermore, pre-existing antibodies to self-antigens have been implicated in increased PGD risk [114,115].

## 6. Structural Consequences of LIRI

### 6.1. Cell Death Mechanisms

Various mechanisms of cell death, including apoptosis, necroptosis, pyroptosis, and ferroptosis, have been implicated in pre-clinical and clinical LIRI [116]. However, the clinical relevance of each cell death mechanism in the context of PGD is not completely understood. Cell-free DNA (cfDNA) can be released from dying cells and may therefore serve as a biomarker for the amount of inflammatory cell death and potentially for donor lung injury. Kanou et al. found that levels of cfDNA in EVLP perfusate were higher in the severe PGD (PGD3) group [117]. Other circulating cell death markers measured during EVLP, including M30, a marker for epithelial apoptosis, and HMGB1 correlated with PGD post-transplant [77]. Similarly, both M30 and M65, which reflect total cell death of epithelial cells, were significantly higher in plasma samples of lung transplant recipients at 24 and 48 h after transplantation [118].

Apoptosis, the most well-known form of programmed cell death (PCD), has been observed in human lungs postreperfusion [119]. Both external stimuli, such as TNF-α released during inflammatory responses, and intrinsic stimuli, including oxidative stress, are able to trigger apoptosis. Triggering apoptosis leads to downstream activation of caspases, colloquially known as the cell’s executioners. Since the cell membrane is not disrupted, this form of cell death has a low inflammatory response [120]. Quadri et al. showed that caspase inhibition in a single-lung transplant model in rats improved lung function. Upon reperfusion, a significant amount of apoptotic cell death was observed in the control group, which was eliminated by caspase 3, 8, and 9 inhibition. This study underlines the potential clinical relevance of apoptosis in the context of lung transplantation. These caspases have also been implicated in the regulation of pyroptosis, yet the potential contribution of this cell death mechanism to the observed improvement in lung function with caspase inhibition was not investigated in that study [121,122]. The occurrence of apoptosis in human lung allografts has been demonstrated [55,119,123]. However, analysis of lung tissue biopsies of 20 donor lungs did not show a significant correlation between the extent of apoptosis and physiologic graft function or short-term outcomes after lung transplant [119]. Apoptosis may contribute to the loss of lung parenchymal cells. However, unlike inflammatory types of cell death, it does not have the capacity to further drive PGD pathogenesis [123].

Necroptosis is another cell death mechanism that has been reported in LIRI [116]. It is characterized by inflammatory downstream effects due to the fact that necroptosis leads to membrane rupture and the release of DAMPs. Activation of necroptosis occurs through phosphorylation of receptor interacting kinases 1 and 3 (RIPK1/3) and subsequent phosphorylation of mixed lineage kinase like (MLKL) [120,124]. Clinical studies investigating the role of necroptosis in PGD are limited. Analysis of human donor lung biopsies demonstrated that mRNA expression of key necroptosis proteins RIPK1 and MLKL were higher in PGD samples [125]. A number of pre-clinical studies have investigated the effect of necrostatin-1, a specific inhibitor of RIPK1, in the context of organ IRI [126,127,128]. In a left lung hilar clamp mouse model, necrostatin-1 treatment significantly alleviated LIRI and inflammatory responses. Necroptosis was primarily found in lung epithelial cells and was inhibited with necrostatin-1. These findings were further evaluated in vitro with a cold ischemia/reperfusion model to mimic the lung transplant setting using immortalized human bronchial epithelial cells with similar results. Inhibition of RIPK1 in this in vitro model not only inhibited necroptosis but also RIPK1-dependent apoptosis. It has been reported that RIPK1 and RIPK3 can mediate apoptosis. Therefore, based on these experiments, the contribution of each cell death pathway cannot be conclusively determined [129,130]. In a rat single-lung transplant model, a left lung was preserved for 18 h at 4 °C and the effect of treatment with necrostatin-1 was evaluated. In the treatment group where both donor lung and recipient were treated with necrostatin-1, lung function was improved through inhibition of RIPK pathway activation. Interestingly, treatment of donor lung or recipient alone was not sufficient to suppress the RIPK pathway [131].

Similar to apoptosis and necroptosis, pyroptosis is a form of PCD frequently implicated in IRI [116,132,133]. Pyroptosis is mediated by inflammasome activation and results in the release of extracellular DNA. Its role in LIRI and potentially the development of PGD is underscored by the pro-inflammatory nature of this type of cell death. Analysis of leukocytes removed during EVLP revealed that a substantial fraction was pyroptotic, indicating that leukocyte pyroptosis may be a mechanism driving tissue injury and contributing to impaired graft function [134].

Although pre-clinical models have demonstrated reduced LIRI by targeting cell death mechanisms, treating cell death alone in the clinical context is likely not sufficient to reduce PGD due to the complexity and potential redundancy of injurious events.

### 6.2. Damage to the Endothelial Membrane

Maintaining endothelial barrier integrity is crucial for preventing the development of PGD. Endothelial dysfunction not only leads to increased permeability but also to the active involvement of ECs in pathological processes such as neutrophil adhesion and coagulation. Cell death is an obvious mechanism by which the endothelial–epithelial barrier is disrupted. In addition, disruption of structural components of the endothelial barrier, including the glycocalyx, extracellular matrix (ECM), and intracellular junctions, results in the formation of gaps between adjacent cells of the endothelium and a pro-inflammatory environment [135].

In LIRI, proteolytic enzymes released by inflammatory cells and parenchymal cells such as neutrophil elastase and matrix metalloproteinases (MMPs) contribute to the degradation of structural components of the endothelium. Oxidative stress can rapidly activate MMPs in the context of organ IRI [136,137,138]. MMPs are primarily known to degrade the ECM components of the basement membrane but have also been reported to proteolyse junctional structures, including VE-cadherin and occludin [139,140]. While a role for MMP activation during an inflammatory response is well described, the exact contribution of MMP activation in the context of PGD is unclear. In a rat lung transplant model, a significant increase in MMP-2 and MMP-9 was observed after only 4 h of ischemia with subsequent reperfusion, and systemic MMP inhibition was found to decrease LIRI [141]. In line with these findings, higher MMP9 expression levels were observed in injured lungs, and inhibition of MMP9 ameliorated LIRI [142]. Elevated plasma MMP levels in lung transplant recipients have been shown to hold prognostic value for bronchiolitis obliterans syndrome (BOS) and chronic lung allograft dysfunction (CLAD), suggesting a long-term effect of MMPs [143,144]. With regard to LIRI, studies investigating MMP effects on short-term outcomes are limited. Nonetheless, research in other solid organs does suggest that MMPs are involved in IRI [138,145,146].

Glycocalyx degradation is another cornerstone in endothelial membrane dysfunction that has received attention in the context of solid organ transplantation [147]. The glycocalyx layer plays a critical role in mitigating oxidative stress, regulating leukocyte–endothelial interactions, and storing macromolecules [148]. One of the key mechanisms for glycocalyx degradation is the activation of proteases such as MMPs, heparanase, and hyaluronidase. In addition, molecules including ROS, thrombin, and plasmin are capable of damaging glycocalyx structures [149]. Processes activated in LIRI induce shedding of the endothelial glycocalyx, exposing pro-thrombotic and pro-inflammatory factors that promote platelet aggregation. Moreover, adhesion molecules that are normally shielded from blood cells by the glycocalyx are exposed, making them available for leukocyte binding [150]. Important constituents of the glycocalyx layer include syndecan-1 and heparin sulfate, and glycocalyx breakdown products have been investigated for their role as biomarkers in sepsis-induced acute lung injury, cardiothoracic surgery, and lung transplantation in human observational studies [151,152,153]. Timothy et al. found that higher levels of donor syndecan-1 levels correlated with the development of PGD grade ≥ 2 in recipients at 24 h post-transplant. In addition, recipient circulating syndecan-1 levels post-transplant remained elevated in PGD grade ≥ 2 compared to those with PGD grade 0–1 [154]. Moreover, an association was found between graft quality and increased levels of glycocalyx degradation products in EVLP perfusate [155]. It is therefore not surprising that researchers have investigated strategies to preserve glycocalyx integrity as a method of preventing LIRI. In a rat lung transplant model, endothelial glycocalyx damage was induced by a warm ischemic insult of 60 min. In some rats, heparin, which is a natural heparanase inhibitor, was introduced prior to the warm ischemic period. After transplantation, glycocalyx degradation was confirmed by ultrastructural changes in lungs that had not received heparin. Additionally, physiological assessment after reperfusion was associated with impaired graft function, inflammation, pulmonary edema, and inflammatory cell migration. In contrast, in lungs treated with heparin, IRI-associated damage was attenuated and the glycocalyx was preserved [156].

### 6.3. Signaling Pathways Altering Vascular Permeability

Another mechanism contributing to endothelial membrane permeability is the combined effect of disruption of cell junctions and cellular contraction. Tight junctions and adherens junctions provide cellular adhesion between neighboring cells and connect to intracellular cytoskeletal elements. In response to inflammatory stimuli, junctional proteins are disrupted. H_2_O_2_ can mediate permeability through ERK1/ERK2 signaling by rearranging the tight junction protein occludin [157,158]. In addition, TNF-α signaling has been shown to disrupt junctional proteins [159]. EC contraction is regulated by intracellular Ca^2+^ signaling and the activation of RhoA, which leads to the phosphorylation and activation of myosin light chain kinase (MLCK) and actomyosin contraction. The increase in cell tension results in gap formation by pulling cells apart, thus making the membrane more permeable. Inflammatory mediators like ROS, thrombin, and VEGF can induce EC permeability through Ca^2+^ signaling. Transient receptor potential channel 6 (TRPC6), highly expressed in human and murine lungs, is activated by oxidative stress and TRPC6-mediated Ca^2+^ entry activates RhoA and its downstream signaling pathway. A mouse model of LIRI demonstrated that ROS production by activated endothelial NADPH oxidase leads to activation of TRPC6 and increased intracellular Ca^2+^, consequently increasing endothelial permeability. Lungs from TRCP6-deficient mice were protected against LIRI after transplantation [160]. In an endothelial cell culture model, signaling through the HMGB1/RAGE axis has been demonstrated to not only induce inflammation via the NF-κB pathway but also directly disrupt endothelial barrier integrity through activation of the RhoA/ROCK1 pathway, which induced gap formation accompanied by the development of stress fiber rearrangement and the disruption of VE-cadherin and ZO-1, crucial proteins in the junctional complexes [161]. A study assessing the association of Rho kinase with LIRI showed that inhalation with an inhibitor in a lung perfusion model with 60 min of warm ischemia protected the lungs against LIRI, demonstrated by decreased lung edema and improved dynamic compliance [162]. Similarly, inhibition of ROCK, a target protein of RhoA, by administering an inhibitor during a donor lung flush 30 min before reperfusion in the recipient rat, attenuated pulmonary edema, the migration of inflammatory cells, and TNF-α production post-transplant [163]. The exact importance of the above mechanisms in PGD has yet to be determined.

## 7. Conclusions

PGD pathophysiology is driven by a complex network of pathways that is shaped by the heterogeneity inherent in the lung transplant process. Cellular and molecular mechanisms of oxidative stress, sterile inflammation, cell death, coagulation, complement activation, and endothelial dysfunction contribute to a loss of integrity of the endothelial–epithelial membrane. In the pre-clinical studies highlighted in this review, targeting specific mediators of injury proved effective in reducing LIRI. However, there is currently no treatment available to treat or mitigate PGD and the only feasible strategies remain protective, such as minimizing warm ischemia and reducing FiO_2_ during donor lung reperfusion. This underscores the need for experimental PGD models, in addition to models for LIRI, that incorporate factors associated with the clinical lung transplantation process, including the use of different donor types (DBD and DCD) and clinically relevant ischemia times and preservation methods. Identifying strategies to assess the extent of donor lung injury prior to transplantation, including biomarker measurement during EVLP, could help predict donor lung suitability. Biomarkers related to inflammation, cell death, and endothelial activation have been identified as promising candidates, and future research should focus on improving the assessment of donor lung quality. In addition, biomarkers could enhance diagnostic precision following transplantation and help guide therapeutic decisions. Capturing the pathophysiology of PGD likely requires the integration of multiple biomarkers that reflect key processes, including neutrophil recruitment, driven by chemokines like CXCL2 and IL-8, and endothelial injury, indicated by markers such as syndecan-1. Focusing solely on pro-inflammatory cytokines (e.g., IL-6, TNF-α, and IL-1β) might not adequately represent lung injury. At present, point-of-care measurement, optimal sampling times, and clinically relevant thresholds remain key challenges, limiting translation into clinical practice. In parallel, the development of multi-targeted therapeutic strategies modulating the complex and redundant mechanisms underlying PGD may hold potential to improve outcomes after lung transplantation, with EVLP providing a valuable platform for such advancements.

## Figures and Tables

**Figure 1 ijms-26-06776-f001:**
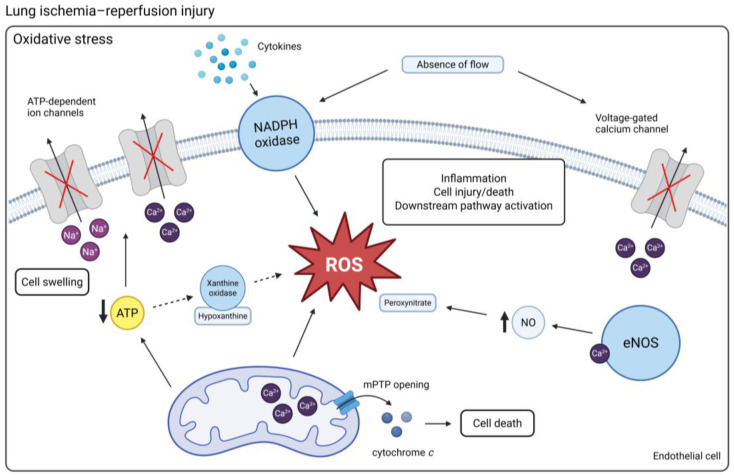
An overview of the mechanisms of oxidative stress in LIRI. Ischemia can result in the depletion of ATP, which inactivates ATP-dependent ion channels. Na^+^ and Ca^2+^ accumulate in the cells, leading to cell swelling. Upon reperfusion, hypoxanthine, a breakdown product of ATP, is converted into superoxide by xanthine oxidase. Activation of NADPH oxidase contributes to the generation of ROS. Increased mitochondrial Ca^2+^ together with the presence of ROS triggers the opening of mPTPs and subsequent cell death. Cessation of flow and associated cell membrane depolarization lead to the inactivation of voltage-gated Ca^2+^ channels, contributing to intracellular Ca^2+^ retention. This stimulates the production of NO by eNOS, which reacts with ROS to form peroxynitrate. (Created in BioRender. Jennekens, J. (2025) https://BioRender.com/6et2zpu). Abbreviations: Ca^2+^ = calcium; mPTP = mitochondrial permeability transition pore; Na^+^ = sodium; eNOS = endothelial nitric oxide synthase; NO = nitric oxide; ROS = reactive oxygen species.

**Figure 2 ijms-26-06776-f002:**
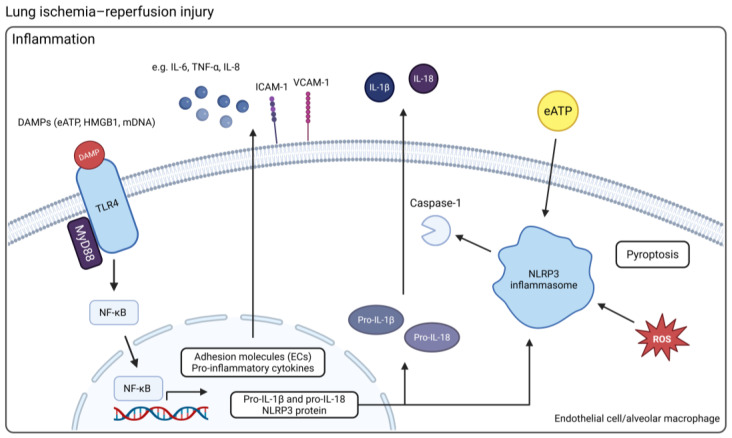
An overview of the mechanisms of innate immune activation in LIRI. DAMPs released by injured or dying cells can activate TLR4, which uses MyD88 to trigger the translocation of NF-κB from the cytosol into the nucleus where it activates the transcription of genes coding for pro-inflammatory cytokines and adhesion molecules in ECs. The NLRP3 inflammasome requires two specific activation signals. First, activation of NF-κB through TLR, NLR, or pro-inflammatory cytokine signaling and downstream upregulation of NLRP3, pro-IL-1β, and pro-IL-18 serves as the priming signal. Next, interaction between NLRP3 and a DAMP, such as eATP or ROS, is required, which results in activation of caspase-1 and cleavage of pro-IL-1β and pro-IL-18 into their mature forms (Created in BioRender. Jennekens, J. (2025) https://BioRender.com/h9my2qg). Abbreviations: DAMP = damage-associated molecular pattern; ECs = endothelial cells; NLRP3 = NOD-like receptor protein 3; eATP = extracellular ATP; ROS = reactive oxygen species. TLR4 = toll-like receptor 4.

**Figure 3 ijms-26-06776-f003:**
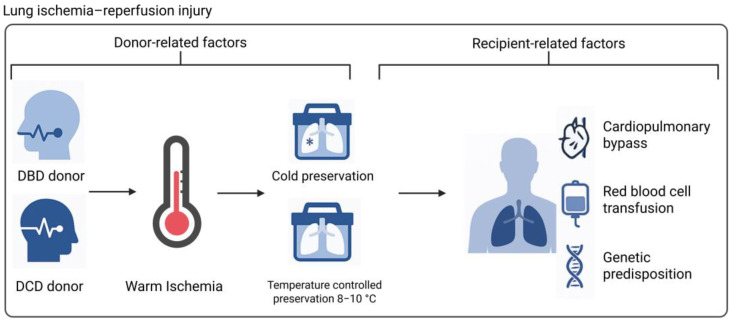
Donor-, procedure-, and recipient-related factors influencing PGD pathophysiology. Donor type, warm ischemic time, preservation method, the use of cardiopulmonary bypass, red blood cell transfusion, and genetic predisposition of a recipient can contribute to donor lung injury prior to retrieval, during preservation, or upon reperfusion (Created in BioRender. de Heer, L. (2025) https://BioRender.com/9fkosgl). Abbreviations: DBD = donation after brain death; DCD = donation after circulatory death.

**Table 1 ijms-26-06776-t001:** PGD grade according to the 2005 International Society for Heart and Lung Transplantation Primary Graft Dysfunction Definition [1].

Grade	Pulmonary Edema on Chest X-Ray	PaO_2_/FiO_2_ Ratio (mmHg)
PGD grade 0	No	>300
PGD grade 1	Yes	>300
PGD grade 2	Yes	200–300
PGD grade 3	Yes	<200

Abbreviations: PaO_2_/FiO_2_ = the ratio of the partial pressure of oxygen in arterial blood (PaO_2_) at a measured fraction of inspired oxygen (FiO_2_).

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
