# Peer review of "Primary Graft Dysfunction in Lung Transplantation: An Overview of the Molecular Mechanisms"

_ijms, 2025, doi:10.3390/ijms26146776_

Round 1

Reviewer 1 Report

Comments and Suggestions for Authors

Several reviews have already been written on the topic of the mechanisms of primary graft dysfunction after allogeneic lung transplantation (DOI: 10.1097/MOT.0000000000001065, DOI: 10.1055/s-0041-1728794, DOI: 10.1080/17476348.2017.1280398, DOI: 10.1097/TP.0000000000004503; DOI: 10.1055/s-0041-1728794; etc.). This review is distinguished by a more detailed description of the molecular and cellular mechanisms of lung graft injury. There are no substantive comments.  The only issue is the title, 'Primary Graft Dysfunction in Lung Transplantation: From Molecular Mechanisms to Clinical Phenotype) is not quite clear to me – what phenotype do the authors mean? Thus, changing the title of the article is not a fundamental requirement, but it could be done.

Author Response

Comment 1: There are no substantive comments.  The only issue is the title, 'Primary Graft Dysfunction in Lung Transplantation: From Molecular Mechanisms to Clinical Phenotype) is not quite clear to me – what phenotype do the authors mean? Thus, changing the title of the article is not a fundamental requirement, but it could be done.

Response 1: Thank you for reviewing our manuscript. We agree with the comment about the title and decided to change the title. The new title is incorporated in the revised manuscript en the title is: Primary Graft Dysfunction in Lung Transplantation: An Overview of Molecular Mechanisms.

Reviewer 2 Report

Comments and Suggestions for Authors

The authors submitted a comprehensive review on the immunopathogenesis of primary graft dysfunction in lung transplantation. The topics is extensively reviewed, the structure of the manuscript is logical, figures are usefull and informative. The manuscript can be considered for publication provided several important issues are addressed:

  1. The author state that Table 1. describes PGD grade according to the 2016 International Society for Heart and Lung Transplantation Primary Graft Dysfunction Definition, reference number 1 in the manuscript, Snell et al, 2017.

There are two important issues regarding Table 1:   

  1. The content of the table differs from the table in Snell et al, 2017 in column 3, second row, the value for is cited as „Any“ in the submitted manuscript (unlike in Snell et al, 2017 where it is >300 mmHg). Please provide an explanation for this difference.
  2. The tables in the reference Snell et al, 2017 and the submitted manuscript are identical (except that one value) and it would be appropriate to make it clear that the table is directly copied from Snell et al, 2017 and not „modified from.....“ (copywright issues etc.).
  3. The title of the manuscript implies that the review deals not only with molecular mechanisms but also with clinical phenotypes. However, the manuscript does not provide sufficient information on clinical phenotypes of patients. Clinical phenotypes in specific diseases are usually generated by integration of research data into models by using deep learning or AI algorithms. It would be useful if the authors could briefly address this issue in the manuscript that carries this specific title (suggested reference but by no means obligatory would be Ronen L, Keshavjee S, Sage AT. Advancing lung transplantation through machine learning and artificial intelligence. Curr Opin Pulm Med. 2025 Jul 1;31(4):381-386. doi: 10.1097/MCP.0000000000001168.). Alternatively, modification of the title is also acceptable.
  4. In section conclusion, it would be helpful if the authors could add a section on their critical interpretation of specific biomarkers of primary graft dysfunction after lung transplantation and identify biomarkers with the highest clinical potential (perhaps in the context of other interpretations such as Feng et al, Front Physiol, 2025)

Author Response

Comment 1:

The author state that Table 1. describes PGD grade according to the 2016 International Society for Heart and Lung Transplantation Primary Graft Dysfunction Definition, reference number 1 in the manuscript, Snell et al, 2017. There are two important issues regarding Table 1:   

The content of the table differs from the table in Snell et al, 2017 in column 3, second row, the value for is cited as „Any“ in the submitted manuscript (unlike in Snell et al, 2017 where it is >300 mmHg). Please provide an explanation for this difference.

Response 1: 

We would like to thank the reviewer for their attentiveness. This was an oversight, and we have corrected column 3, row 2 to >300 mmHg.

Comment 2:

The tables in the reference Snell et al, 2017 and the submitted manuscript are identical (except that one value) and it would be appropriate to make it clear that the table is directly copied from Snell et al, 2017 and not „modified from.....“ (copywright issues etc.).

Response 2:

In fact, the table is based on a publication from the ISHLT in 2005, but it was not copied verbatim. Therefore, we have updated the reference accordingly in the revised document (line 48 and lines 615 to 617).

Comment 3:

The title of the manuscript implies that the review deals not only with molecular mechanisms but also with clinical phenotypes. However, the manuscript does not provide sufficient information on clinical phenotypes of patients. Clinical phenotypes in specific diseases are usually generated by integration of research data into models by using deep learning or AI algorithms. It would be useful if the authors could briefly address this issue in the manuscript that carries this specific title (suggested reference but by no means obligatory would be Ronen L, Keshavjee S, Sage AT. Advancing lung transplantation through machine learning and artificial intelligence. Curr Opin Pulm Med. 2025 Jul 1;31(4):381-386. doi: 10.1097/MCP.0000000000001168.). Alternatively, modification of the title is also acceptable.

Response 3:

As advised by another reviewer, we have decided to revise the title to: Primary Graft Dysfunction in Lung Transplantation: An Overview of Molecular Mechanisms.

Comment 4:

In section conclusion, it would be helpful if the authors could add a section on their critical interpretation of specific biomarkers of primary graft dysfunction after lung transplantation and identify biomarkers with the highest clinical potential (perhaps in the context of other interpretations such as Feng et al, Front Physiol, 2025)

Response 4:

We agree that a section on critical interpretation of biomarkers was lacking in the conclusion. In the revised document we have added a paragraph addressing this topic. In addition to the study by Feng et al, which emphasizes biomarkers to diagnose PGD post-transplant, the study by Costamagna et al, Transplant International, 2025 focusses on biomarkers during EVLP. Both studies have been reviewed to formulate our interpretation (lines 602-609).

Round 2

Reviewer 2 Report

Comments and Suggestions for Authors

The authors corrected everything according to suggestions. The manuscript is recommended for publication in the present form.